# Creation and Implementation of a New Sentinel Surveillance Model in Pharmacy Offices in Southern Europe

**DOI:** 10.3390/ijerph19148600

**Published:** 2022-07-14

**Authors:** Anna M. Jambrina, Neus Rams, Pilar Rius, Maria Perelló, Montserrat Gironès, Clara Pareja, Francisco José Pérez-Cano, Àngels Franch, Manel Rabanal

**Affiliations:** 1Directorate-General for Healthcare Planning and Regulation, Ministry of Health, Government of Catalonia, 08028 Barcelona, Spain; n.rams@gencat.cat (N.R.); cpareja@gencat.cat (C.P.); mrabanal@gencat.cat (M.R.); 2Physiology Section, Department of Biochemistry and Physiology, Faculty of Pharmacy and Food Science, University of Barcelona, 08028 Barcelona, Spain; mperello@ccfc.cat (M.P.); franciscoperez@ub.edu (F.J.P.-C.); angelsfranch@ub.edu (À.F.); 3Council of the Pharmacist’s Association of Catalonia, 08009 Barcelona, Spain; prius@ccfc.cat (P.R.); mgirones@ccfc.cat (M.G.); 4Institute of Research in Nutrition and Food Safety (INSA), 08921 Santa Coloma de Gramenet, Spain

**Keywords:** pharmacy, health services administration, pharmacist intervention, pharmacies sentinel network, public health, community health, community pharmacies

## Abstract

Traditionally, health sentinel networks have focused on the reporting of data by primary care physicians and hospitals, ignoring the role of the community pharmacist as an expert in drugs. The objective of this study was to describe a method for creating a network of sentinel pharmacies in a region of Southern Europe in order to have a pharmaceutical surveillance system that is representative of the territory to be monitored and that can respond to any events or incidents that can be followed up by the community pharmacy. The creation process was carried out in three phases: a first phase of selection through a cluster and population analysis and a final adjustment, a second phase of voluntariness and random selection, and a third phase of training and implementation of the network. A sentinel network of 75 community pharmacies has been established in Catalonia. The network monitors 2.47% of the total population with a homogeneous proportion of urban (42), rural (30), and mountain-area (3) pharmacies based on the particular characteristics of the territory. This model allows increased surveillance in the territory, objectively and representatively detects problems arising from the use of medicines, and establishes improvement strategies of public health.

## 1. Introduction

A health sentinel network is defined as an information system aimed at public health surveillance and epidemiological research based on the voluntary collaboration of health professionals to study the frequency of diseases and the determinants of health [1,2].

Over time, sentinel networks have gained more and more prominence in Europe as well as in many other countries [2,3,4,5]. In Europe, the first sentinel network emerged in the United Kingdom and is dated to the mid-20th century. Later, Belgium, France, and the Netherlands incorporated more developed surveillance systems [2,3,6].

The initial objectives of health sentinel networks have evolved over the last few years, adopting new working methods and using new tools for their development. The change in the epidemiological pattern of the diseases that cause the greatest impact on the population, mainly associated with life habits and aging, have expanded the area of action of this information system to chronic and degenerative processes and their risk factors, what is commonly known as ‘public health surveillance’ [1]. This surveillance is defined as the systematic collection, analysis, and interpretation of data for health purposes as well as its dissemination, with the aim of reducing morbidity and mortality [7]. Once collected, these data are essential to determining whether a preventive intervention is having any effect on the health of the population and also to evaluate if new strategies requiring an intervention and more exhaustive follow-up are necessary.

The surveillance of chronic diseases is one of the most pressing challenges for health systems worldwide because they are the main cause of death and disability across the globe. This calls for multidisciplinary health participation. In Spain, sentinel systems have historically focused on the reporting of data by groups of voluntary health workers, mainly physicians and pediatricians from primary care and hospitals [6]. An emerging consensus among academics, professional organizations, and policymakers determined that community pharmacists should take on a broader role in contributing to the safe, effective, and efficient use of medicines, especially in people with multiple chronic conditions [8]. Community pharmacists can help to improve health by reducing medication-related adverse events and by educating the public on the rational and safe use of medicines. Likewise, they can help to achieve better medication adherence, which in turn can reduce unnecessary provider visits, hospitalizations, and re-admissions while improving the provision of integrated primary care throughout the health system [8,9].

It should be noted that the demand for health care through less complex processes is met mainly in the community pharmacy since it is the first level of health care. Consequently, the pharmacist has an enormous potential, offering the possibility of monitoring the demand for pharmaceutical indications related to the preclinical stages of pathological processes. Furthermore, they also have the responsibility of reporting about the rational use of medicines and educating the population to adopt safe behaviors.

In Catalonia, a region in Southern Europe, the powers of community pharmacists are regulated by the Pharmaceutical Regulation Law. This Law establishes that pharmacy offices will collaborate in programs promoted by the Health Administration or the pharmaceutical association on quality assurance in pharmaceutical care and in health care in general, health promotion and protection, disease prevention, and health education [10].

For all of the above, in 2017, the Catalan Ministry of Health and the Council of Colleges of Pharmacists created a network of sentinel pharmacies in Catalonia. In it, community pharmacists collaborate with the Health Administration in the observation, detection, and reporting of events of interest related to the surveillance of the prescription, dispensing, and administration of medicines as well as in the description of population-level behaviors regarding important health problems.

During the 2017–2021 period, the network of sentinel pharmacies in Catalonia facilitated the effectiveness and efficiency of traditional surveillance programs by obtaining valid data related to the safety of medicines generated by the dynamics of health care in the community pharmacy office. Furthermore, it made it possible to identify individual and population-level behaviors and to provide an overall view of the burden of the disease and the risks associated with the use of medication [11,12,13]. Likewise, this five-year study process allowed the optimization of the selection method of community pharmacies based on the experience gained in the studies that had been carried out.

Therefore, the objective of this study was to describe a method for creating a network of sentinel pharmacies in order to have a pharmaceutical surveillance system that is representative of the territory to be monitored and is able to respond to any health-related phenomenon that can be surveilled or monitored through the community pharmacy.

## 2. Materials and Methods

The creation of a network of sentinel pharmacies was carried out in three phases: A first phase of selection, a second phase of voluntariness and random selection, and a third phase of training and implementation of the sentinel network.

### 2.1. Selection Phase

The objective was to determine the minimum number of sentinel pharmacies and the location of these to obtain the greatest possible representativeness. To do this, the following procedures were used:

#### 2.1.1. Cluster Analysis

First, the selected pharmacies had to ensure the representativeness of the population to be monitored, so stratified sampling was conducted in which conglomerates of health zones defined the strata. The health zones were grouped in an appropriate proportion of urban, rural, and mountain-area strata.

#### 2.1.2. Population Analysis

The number of pharmacies was conditioned by the size of the minimum population to be monitored and the variables and indicators to be studied as well as the management and operation capacity of the sentinel network. To this end, other existing sentinel network models were considered in which it is assumed that good surveillance must take into account the maximum admissible error. This variable is defined as the percentage of variation of the standard error with respect to the proportion that is estimated, with a confidence interval of 95%, together with the minimum number of reports to accurately describe the variable to be monitored [1]. In this respect, for most medium-frequency problems monitored by health sentinel networks in Spain, experience in calculating the samples of other sentinel networks determined a minimum of 50,000 inhabitants for health indicators [1]. Given that a larger sample makes it possible to monitor low-frequency processes using sample calculation methods already described in the scientific literature, it was established that monitoring at least 150,000 inhabitants would allow data with a high value of representativeness to be obtained [1]. 

#### 2.1.3. Adjustment for Strata and Population

Once the previous procedures had been carried out, the proportion of pharmacies based on urban, rural, and mountain-area strata and the percentage of the population to be monitored in each zone were available. Taking these two factors into account, each of the health regions was reviewed to establish the minimum number of pharmacies that met both criteria. In the event that due to either of the two factors the number of pharmacies was high, these must be distributed homogeneously across the basic health areas (BHA) and the health sectors of each health region. 

### 2.2. Voluntariness and Random Selection Phase

The voluntary involvement of the pharmacy is one of the additional characteristics of this network that mitigates bias in the reporting depending on the degree of motivation of the pharmacist. 

### 2.3. Training and Implementation Phase

An essential aspect when starting up the sentinel network is that all pharmacies have the same information, in order to standardize data collection. For this reason, it is necessary to develop a training program from which pharmacies can start their activity. 

## 3. Results

### 3.1. Cluster Analysis

In Catalonia, community pharmacies are health facilities subject to planning that is specified by local health regulations, which draw up a health map defined by geographic, economic, social, and demographic factors (Figure 1). 

The health map establishes that Catalonia has seven health regions (HR) (Figure 1A). These HR are, in turn, divided into 29 health sectors (HS) (Figure 1B), which group the 374 basic health areas (BHA) (Figure 1C), which can be classified into urban (uBHA), rural (rBHA), and mountain areas (mBHA). The distribution of community pharmacies is based on the BHA. Therefore, the result of this first analysis allowed us to obtain the proportion of urban, rural, and mountain-area pharmacies in each HR in order to derive a homogeneous distribution for urban, rural, and mountain-area strata.

Table 1 shows the results obtained with this analysis. For example, the Alt Pirineu i Aran HR is made up of two HS. These HS group eight BHA (four rBHA and four mBHA). For this, the proportion of pharmacies established in this HR is zero urban, one rural, and one mountain. This analysis was carried out with the seven HR in Catalonia.

### 3.2. Population Analysis

Catalonia has 3185 community pharmacies, serving an average of 2500 people each. Given that it was established that monitoring at least 150,000 inhabitants would allow to data with a high value of representativeness to be obtained and taking into account that, at the time of the creation of the network of sentinel pharmacies, Catalonia had a population of 7,600,069 inhabitants, this meant monitoring at least 2% of the population of Catalonia, a parameter that must be met by each of the previously selected conglomerates.

Table 2 shows the analysis carried out based only on the demographic factors of each HR. The results derived from this analysis show that at least 63 community pharmacies would be necessary to monitor at least 2% of the population.

### 3.3. Adjustment for Strata and Population

Table 3 shows the number of pharmacies required in each HR based on the final adjustments by strata and population. The adjustment takes into account two main criteria. On the one hand, it uses the total number of pharmacies that would be necessary in view of the analysis by conglomerates and population. On the other hand, the special characteristics of each HR to ensure 2% of the population is monitored in each of the types of region (urban, rural, and mountain-area) was also required.

With this adjustment, the final number of community pharmacies in the Catalan sentinel pharmacy network was 75: 42 urban, 30 rural, and 3 mountain-area pharmacies. This represents approximately that 2.47% of the total population of Catalonia being monitored.

Finally, the pharmacies were selected from the municipalities where the average number of inhabitants per pharmacy was representative of the territory since there may be pharmacies in municipalities with few inhabitants or pharmacies with a catchment area including a much larger population than the average for that stratum.

### 3.4. Voluntariness and Random Selection Phase

Once the number of community pharmacies necessary to ensure the exhaustiveness and validity of the information was determined, these pharmacies were invited to take part on a voluntary basis.

The Official Colleges of Pharmacists of Catalonia sent a formal invitation for participation to all the community pharmacies in each of the selected municipalities, and the community pharmacies interested in participating in the network voluntarily registered over a fortnight. Likewise, during the registration period for the pharmacies, an informative session was held to explain the objectives and the development of the network and to answer all related questions. 

For geographic areas with more than one pharmacy interested in participating, a random selection was made to determine which pharmacy took part. This random selection (Figure 2) was carried out by means of a public draw among the pharmacies involved with the presence of representatives of the Catalan Ministry of Health and the Council of Colleges of Pharmacists of Catalonia, who are the health institutions who lead, coordinate, and manage the sentinel network.

### 3.5. Training and Implementation Phase

The sentinel pharmacies selected to participate in the network participated in a specific training program to ensure they had the knowledge, attitudes, and skills necessary to carry out the surveillance activities. The training session was held through a theoretical–practical workshop where experts in each of the activities presented the topics to be discussed, the objective of the monitoring, and how to record all the information in the electronic forms prepared for the purpose. This workshop lasted five hours, and 100% of the selected pharmacies attended the training.

Work was carried out in three large areas of intervention:

#### 3.5.1. Pharmacovigilance

This step was implemented in order to provide additional information on medication errors and suspected adverse drug reactions as well as to monitor certain groups of drugs to improve risk-minimization strategies. In relation to the selection of activities, it is very important to have quality indicators for activities that allow the comprehensive assessment of information. 

#### 3.5.2. The Creation of an Observatory of Drugs of Abuse

This was created with the aim of detecting and reporting the abuse or recreational use of drugs in order to implement preventive measures [11].

#### 3.5.3. Public Health Activities

These actions were implemented with the purpose of providing information supplementing that recorded by traditional surveillance systems. In this respect, the sentinel pharmacies of Catalonia have participated in projects such as obtaining daily information on acute respiratory diseases. The aim was detecting the start of the annual flu epidemic in advance and monitoring the dispensing of contraception with a goal of obtaining information on the profiles of users and other conditions that may influence and provide criteria to evaluate and/or improve the Public Health Emergency Contraception Program’s strategies [12,13].

Table 4 shows the activities carried out during the five years of study.

For the collection and analysis of the information, the community pharmacists had to complete validated questionnaires adapted to each of the activities and send them through an online registration system to the coordination units responsible for their analysis, thus ensuring data confidentiality and protection. Table 5 shows a sample questionnaire for each of the intervention areas. It is important to have a “control activity” from which the exact number of reports that should be received is known. In this case, the return of monitoring of pharmaceutical quality alerts was chosen as the control activity.

Detailed results of some of these activities have been published in international scientific journals [11,12,13]. Others are still under study. However, the main conclusions drawn from them are as follows:Monitoring of pharmaceutical quality alerts: only 8.1% of the quality alerts issued affect batches of medicines that are dispensed in community pharmacies.Notification of medication errors: in 69.4% of cases, the pharmacist was able to avoid the medication error.Notification of suspected adverse drug reactions: the therapeutic groups involved in most suspected adverse reactions are painkillers and anti-inflammatory drugs (17%).Additional follow-up of treatment with SGLT2 inhibitors: a total of 77 suspected adverse reactions have been reported, most related to dapagliflozin (51.9%).Monitoring of the dispensing of valproic acid in adult women of gestational age: there is a significant ignorance of teratogenic risks in patients treated with valproic acid (54.9%).Monitoring of the dispensing of isotretinoin in adult women of gestational age: most women on isotretinoin treatment were aware of the teratogenic risks associated with its use (75%), but in practically half of the cases, women did not receive the informative material oriented to the prevention and pregnancy control measures.Monitoring of abuse or recreational use of medications: the drugs most frequently implicated in abusive behaviors are benzodiazepines (36.3%), mainly in young men [11].Use of tramadol for non-oncologic pain: the majority (60%) of patients have needed to increase the dose of tramadol without consulting a doctor, and 13.8% have used it for symptoms other than pain.Characterization of the use of emergency contraception: many users (44.2%) said they had taken the medication on other occasions, and the number of women who requested the medication when they were recurring was higher than the first users (85.8% vs. 75.4%) [13].Influenza syndromic surveillance: sentinel pharmacies make it possible to detect the start of the flu epidemic a week earlier than health centers [12].

Likewise, systems for returning information to the participants must be available in order to obtain greater adherence to the network and safeguard the quality of data. To this end, semi-annual reports were prepared with the results of each of the activities and each pharmacy’s participation in the project. Two annual follow-up meetings were also scheduled with the pharmacists of the sentinel pharmacies to show the obtained results, resolve queries, and determine future strategic lines and/or action measures aimed at improving the behaviors and habits of the population. 

All this was complemented with specific workshops to improve the knowledge, skills, and attitudes of the professionals involved in the project. During the 2017–2021 period, the following activities were developed annually:A workshop of medication error detection (duration: 4 h).A course on the detection of suspected adverse drug reactions (duration: 3 h).

Both activities were developed in two parts. Initially, a theoretical part was taught to improve the knowledge of pharmacists in these subjects. Next, real cases of received notifications were used, with the aim of improving the information collected and the ability to detect cases.

Similarly, monthly newsletters were prepared that served as a vehicle for relevant information related to the main results of each project’s activities and were sent to the pharmacy offices that participated in the network. During the five years of the study, 60 newsletters were published. The topics covered in these newsletters have focused mainly on:Periodic summaries of the results obtained in each activity.Review articles on possible drug abuse trends that were likely to be monitored.Information on participation in scientific publications and conferences.Warnings or information on aspects of improvement that needed to be reinforced in each of the activities.

## 4. Discussion

The importance of public health surveillance lies in the planning, implementation, and evaluation of health strategies and the integration of data in the decision-making process to help prevent diseases or health disorders in the population [7]. Sentinel networks are a key piece in this surveillance, as are health professionals on the frontline. However, contrary to what happens with other health professionals, the expansion of the role of community pharmacists participating in health policies has always been limited [14]. Despite their considerable training, community pharmacists are the only health professionals who are not primarily rewarded for providing health care. For this reason, they are underutilized as public health professionals, even though they are the third largest health professional provider group in the world after physicians and nurses [8].

To date, existing sentinel models have focused on surveillance networks based on the voluntary participation of health professionals, mainly primary care physicians, who report cases of acute illness during routine clinical practice. The most common are present in many countries, for example, the United States Influenza-Like Illness Surveillance (ILINet) and the European Influenza Surveillance Network (EISN) in 29 European countries [15,16]. Nevertheless, most require further optimization of the selection of data providers and an improvement in incidence estimates, together with better representativeness of the monitored population in terms of demographic and socioeconomic characteristics and the possibility of selecting participants, all with a view to ensure that the estimates of the surveillance network are as close as possible to a reference epidemiological signal [17,18].

Up to now, there have been no precise methods for determining pharmacy-level characteristics that support high levels of pharmacist performance and that can provide guidelines for improving pharmacy policy. Studies have evaluated pharmacy characteristics, such as workload, continuity of care, culture, and workflow and staffing, individually and with various definitions of quality performance. The few studies that used standardized quality measures used a potentially biased ecological approach to estimate pharmacy characteristics. In addition, they use to determine a population-based quality metric in the geographic area and then assign these population-based results to all pharmacies within that geographic area. However, more robust methodologies are needed to measure patient characteristics and pharmacy and workload status when the patient receives medication [9].

In Europe, there are four different pharmacy office systems in place. The Scandinavian-type pharmacy has relatively large pharmacies, serving 10,000–18,000 people and focusing mainly on medicine. Southern Europe, France, and Belgium have very small pharmacies that serve approximately 2000–2500 people and also sell OTC drugs and cosmetics. The United Kingdom and Israel follow the English-speaking model (United States and Australia), focused on the sale of non-medical items in addition to medicines and serving a population of approximately 3500 users. Finally, there are the pharmacies in Central and Eastern Europe (Germany, Switzerland, and Austria) which focus on all types of health services and serve 3000–5000 people.

In Catalonia, the Southern Europe model is in place and, in this respect, the Catalan community pharmacy has specific attributes that are strengths in terms of strategic health planning. The most significant is the fact that it is considered the first point of health care and, therefore, where the demand for care of low or very low complexity health processes is to be found. It is a dense network with a multitude of service points that is highly accessible and of great proximity and social capillarity, which gives it a degree of equity across the territory greater than that of any other health device. At the head of these centers, there is always a lead pharmacist available for users [19,20].

Despite the principle of all sentinel networks being the same, the variations regarding their objectives, communication systems, financing, and organizational and functional dependence may vary considerably. Although there was not this high degree of variability in Spain, it was necessary to create a guide of principles and methods. With this, it could be possible to harmonize all the health sentinel networks in Spain under similar criteria, thus enabling reliable inter-regional comparisons and conducting studies coordinated across various networks. Furthermore, there was a demand for a common model that could be easily rolled out in our healthcare system. For all these reasons, in 2006 the “Guide to the principles and methods of health sentinel networks” was drawn up as part of the RECENT project. This project steered the creation of this new sentinel pharmacy network model in Catalonia, which is made up solely of community pharmacy professionals [1,2].

The methodology for creating this network of sentinel pharmacies combines two fundamental aspects: flexibility and specificity. Flexibility implies allowing adaptation to different environments and situations depending on the characteristics of the territory. It also refers to carrying out the surveillance of various activities based on unmet health needs. Specificity is fundamental because the ambiguity in this system can give rise to methodological drift that distorts it. Hence, in this case, the methodology of the network was based on a need that had not been met until now. In this case, it allows for the integration of the community pharmacist professionals and demonstrates that they are well-prepared to assume a broader role in the health system if granted expanded responsibilities in caring for people with multiple chronic conditions.

In Spain, there are other pharmacy sentinel surveillance network projects in other regions such as Castilla y León, Madrid, and Navarra, the latter being recently created. All of them have the objective of the detection, reporting, and prevention of safety problems related to the use of medicines such as adverse reactions or medication errors [21]. Unlike previous programs, the Catalan sentinel pharmacy network model has been created following the “Guide to the principles and methods of health sentinel networks”. This approach means that the community pharmacy is integrated into the territory and has a fundamental impact on the representativeness of the reference population and in the possible biases in the estimations of indicators [1,2]. This has allowed for the creation of a robust model for conducting research projects that contribute to the safer use of medicines. In addition, this would allow the corroboration of the importance of the pharmacovigilance tasks of the community pharmacist to contribute to the well-being of patients and the improvement in public health.

It has to be considered that this sentinel pharmacy network model studies and estimates very different variables and indicators with a great diversity in the population variances. For this reason, the statistical approximation of the ideal size of the sample relative to the number of participating pharmacies was difficult to determine, and it required a final adjustment that would consider the special features of the territory in terms of geography and population density.

However, this new sentinel network model has many strengths that make it distinctive. On the one hand, it is the first model that has as its key player only the community pharmacy and the community pharmacist as a professional specialized in drugs and with great proximity and accessibility to the patient. On the other hand, being a dynamic and cross-cutting project, the monitoring of various activities focused on drug surveillance and patient safety stands out. Finally, it is not a closed project, but rather it can and should be modified depending on the obtained results. All this represents a benefit in the field of the standardization of health indicators internationally and greater visibility for and professionalization of the community pharmacy model.

## 5. Conclusions

The importance of preventing safety problems related to the use of medicines is mainly addressed by pharmacovigilance and patient education tasks that health professionals carry out in their daily care practice. For this reason, the proximity characteristic of community pharmacists is of importance (because of their closeness to patients and their knowledge of drugs). It makes it possible to add to the information obtained from already established traditional surveillance systems. At the same time, this relationship of closeness and trust with patients helps the early detection of inappropriate behaviors related to the use of medicines, thereby preventing major health problems and protecting the population. The creation of this new sentinel pharmacy network model in Catalonia allows for increased surveillance in the territory. This occurs due to the advantage of the health and professional resources made available by the pharmacist to objectively and representatively detect problems arising from the use of medicines and to establish improvement strategies to prevent them.

## Figures and Tables

**Figure 1 ijerph-19-08600-f001:**
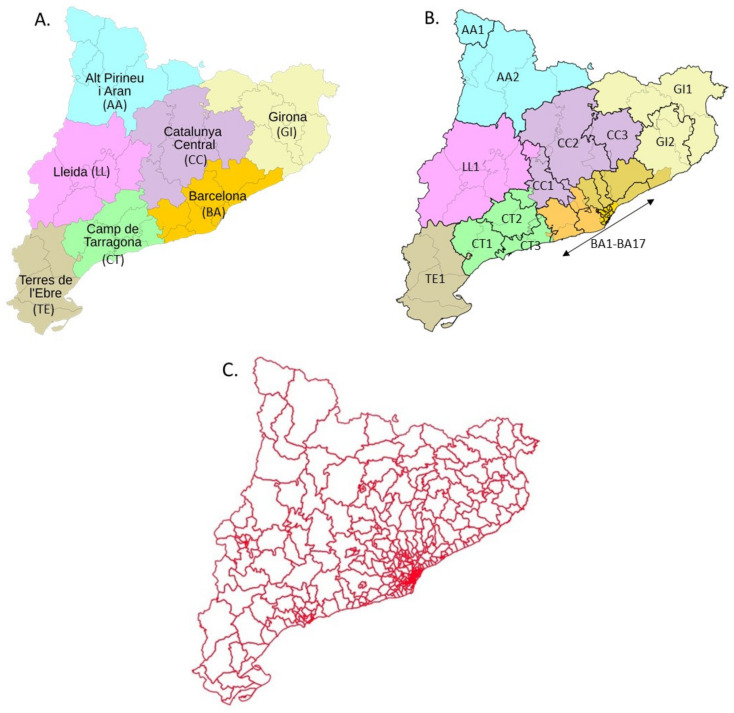
Health map of Catalonia distributed into health regions (**A**), health sectors (**B**), and basic health areas (**C**). The names of the health regions are included in panel (**A**) and the codes of both the health regions and the health sectors are also included in panels (**A**) and (**B**), respectively. Health sector codes are based on the health sector root code.

**Figure 2 ijerph-19-08600-f002:**
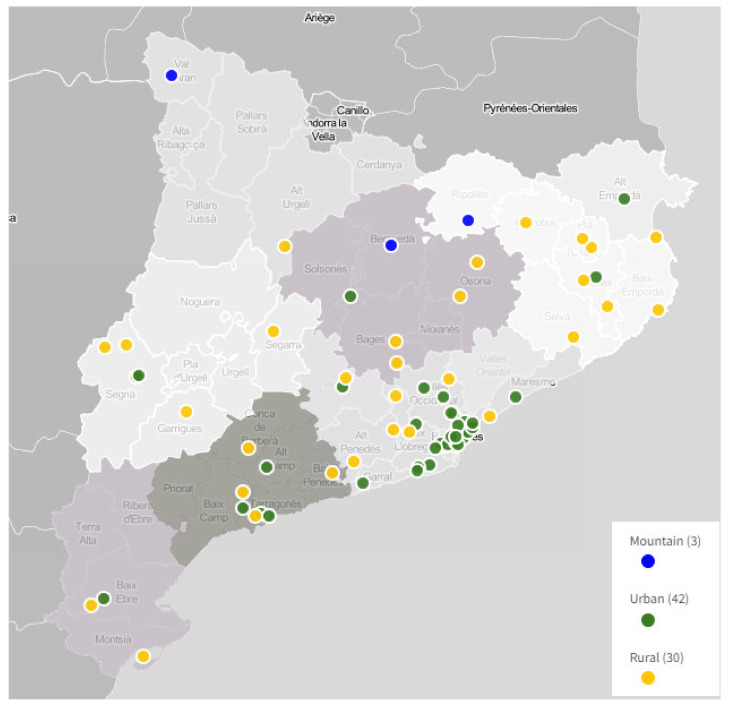
Map of the final territorial distribution of the 75 pharmacy offices of the sentinel pharmacy network of Catalonia.

**Table 1 ijerph-19-08600-t001:** Cluster analysis to obtain the proportion of pharmacies in each HR.

HR	HS (n)	BHA (n)	BHA Type ^1^	Ratio ^2^
Alt Pirineu i Aran	2	8	0:4:4	0:1:1
Lleida	1	23	7:16:0	1:2:0
Camp de Tarragona	3	33	16:17:0	1:1:0
Terres de l’Ebre	1	11	2:9:0	1:4:0
Girona	2	41	7:28:6	1:8:1
Catalunya Central	3	38	11:18:9	1:2:1
Barcelona	17	220	184:36:0	5:1:0

^1^ Number of uBHA, rBHA, and mBHA in each HR. ^2^ Proportion of urban, rural, and mountain-area pharmacies in each HR.

**Table 2 ijerph-19-08600-t002:** Population analysis to obtain the minimum number of pharmacies in each HR.

HR	Population (n)	2% Population Monitored (n)	Pharmacies (n)
Alt Pirineu i Aran	71,888	1438	1
Lleida	361,702	7234	3
Camp de Tarragona	617,504	12,350	5
Terres de l’Ebre	178,398	3568	2
Girona	865,282	17,306	7
Catalunya Central	519,836	10,397	5
Barcelona	4,985,459	99,709	40
**Total**	**7,600,609**	**152,002**	**63**

**Table 3 ijerph-19-08600-t003:** Population analysis to obtain the minimum number of pharmacies per HR.

HR	Ratio CA ^1^	Pharmacies (n) PA ^2^	Pharmacies (n) ^3^	Special Characteristics	Total Pharmacies (n)
Alt Pirineu i Aran	0:1:1	1	2 (0U, 1R, 1M)	None	2 (0U, 1R, 1M)
Lleida	1:2:0	3	3 (1U, 2R, 0M)	Since 1 urban pharmacy did not ensure 2% of the urban population was monitored, the ratio was expanded to 2 urban and 4 rural pharmacies.	6 (2U, 4R, 0M)
Camp de Tarragona	1:1:0	5	6 (3U, 3R, 0M)	Since 3 urban pharmacies did not ensure 2% of the urban population was monitored, it was expanded to 4 urban and 4 rural pharmacies.	8 (4U, 4R, 0M)
Terres de l’Ebre	1:4:0	2	5 (1U, 4R, 0M)	A much larger number of rural than urban inhabitants were monitored, which could lead to a representativeness bias. To equate the two population groups, the number of rural pharmacies was reduced to 2.	3 (1U, 2R, 0M)
Girona	1:8:1	7	10 (1U, 8R, 1M)	Given that 1 urban pharmacy did not ensure 2% of the urban was population monitored and that the ratio allowed for a second urban pharmacy, 2 urban pharmacies were selected.	11 (2U, 8R, 1M)
Catalunya Central	1:2:1	5	5 (1U, 2R, 1M, and 1 uncategorized)	Given that at least 5 pharmacies were needed and that the ratio allowed for a second urban pharmacy, 2 urban pharmacies were selected.	5 (2U, 2R, 1M)
Barcelona	5:1:0	40	40 (32U, 8R, 0M)	None	40 (32U, 8R, 0M)

^1^ CA: Cluster analysis. ^2^ PA: Population analysis. ^3^ Number of pharmacies according to the two analyses (CA and PA) and the number of urban pharmacies (U), rural pharmacies (R), and mountain-area pharmacies (M).

**Table 4 ijerph-19-08600-t004:** Activities carried out for the network of sentinel pharmacies in Catalonia during the 2017–2021 period.

Area of Intervention	Name of Activity	Duration	Collected Notifications (n)
Pharmacovigilance	Monitoring of pharmaceutical quality alerts	5 years	4918
Pharmacovigilance	Notification of medication errors	5 years	1676
Pharmacovigilance	Notification of suspected adverse drug reactions	5 years	582
Pharmacovigilance	Additional follow-up of treatment with SGLT2 inhibitors	1 year	505
Pharmacovigilance	Monitoring of the dispensing of valproic acid in adult women of gestational age	2.5 years	91
Pharmacovigilance	Monitoring of the dispensing of isotretinoin in adult women of gestational age	1 year	98
Observatory of drug abuse	Monitoring of abuse or recreational use of medications [11]	5 years	1229
Observatory of drug abuse	Use of tramadol for non-oncologic pain	2 years	266
Public health	Characterization of the use of emergency contraception [13]	2 years	941
Public health	Influenza syndromic surveillance [12]	5 years	1986

**Table 5 ijerph-19-08600-t005:** Validated questionnaire templates for sentinel pharmacy data collection.

Area of Intervention	Name of Activity	Question Number	Question Text
Pharmacovigilance	Monitoring of pharmaceutical quality alerts	1	Pharmacy ID
		2	Pharmaceutical alert ID
		3	Reception date of the alert
		4	Reception channel of the alert
		5	Has pharmacy staff been informed?
		6	Are batches of the drug affected in your pharmacy?
		7	Date of withdrawal of the batch medicine
Observatory of drug abuse	Monitoring of abuse or recreational use of medications	1	Pharmacy ID
		2	Pharmacist
		3	Patient age
		4	Patient sex
		5	Patient origin (native or not)
		6	Substance involved
		7	Drug request (prescription or not)
		8	Request with or not intimidation
		9	Frequent request?
		10	Pharmacist management?
		11	Why do you supply the medicine?
		12	Observations
Public Health	Characterization of the use of emergency contraception (EC)	1	Pharmacy ID
		2	Data of EC dispensation
		3	National code and name of medicinal product
		4	User sex (male, female)
		5	Age of the user
		6	Whom the medication is for?
		7	Population of residence
		8	Time from unprotected sex (hours)
		9	Time since the last menstrual period (weeks)
		10	Contraceptive method normally used
		11	First EC dispensation or not
		12	Suspected adverse reactions reported?
		13	Description of pharmaceutical action performed
		14	Availability of an EC kit consisting of a condom and additional informative material (yes or not)
		15	Observations

## Data Availability

The datasets that support the findings of this study are available from the first author (M.P.) upon reasonable written request.

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
