# Peer review of "Creation and Implementation of a New Sentinel Surveillance Model in Pharmacy Offices in Southern Europe"

_ijerph, 2022, doi:10.3390/ijerph19148600_

Round 1

Reviewer 1 Report

The topic is very interesting and promising. Pharmacies could be the first contact between the population and the health system. A health sentinel network that builds on a representative sample of pharmacies from an area is a very promising work. A lot of public health indicators, including population drug use measures, could be collected proactively or passively based on such a network of pharmacies. Moreover, good research could be implemented based on such a network. Additionally, training and incentive for pharmacy staff for this are also important, for most the pharmacies have a private-for-profit feature, and is not a part of the public health system sector. I learned from the paper that a training process had been conducted as well.

However, despite the value of the sentinel network, the article per se has much room to improve. The article mainly describes the process of the development of the network, I have the following suggestions.

First, as a research paper, its title, study aims, methods, and results parts are not matching well. The title infers that the authors might have conducted evaluation work to test the validity of the created network, however, I don’t see any description of the methods or results of the evaluation in the current paper.

Second, in my view, the content under subtitle “2.3 Training and implementation phase” in the methods part in the current form is not methodologies, instead, more like activities in practice.

Third, the results corresponding to 2.3 of the methods part, which were presented in 3.5 in the results part, only showed what had been conducted, instead of what had been found.

Forth, the authors did a systematic sampling to try their best to draw a representative sample of pharmacies that can cover the population in Catalonia, which is excellent work. However, the description of this sampling is overlapping between the methods and results part. Need efforts for a better structure.

Reviewer 2 Report

Thank you for giving me the chance to review this interesting article 

I understand that so much work was done to build this network.

In my opinion as a reader, the idea is excellent, but the representation could be improved as you focused mainly on stage 1 so much is written about sampling and geography of the region.

What I feel is missing as an international pharmacist is the results section you mentioned annule reports and workshops, but not much details is written in this section. I would suggest maybe adding some examples of surveys that pharmacists completed for the work to be reproducible and expand the findings with more details.

In general, it is a good manuscript and recommend accepting it after minor corrections  

Here are my specific comments 

1.     Provide more details about the second part of the results workshop, annual reports. Maybe an example in the appendix

2.     Provide an example of surveys pharmacists completed   

Reviewer 3 Report

The manuscript has a clear, logical structure. Figures and tables are clearly presented. The strength and novelty of the paper is the structured involvement of pharmacists in a sentinel network in public health surveillance. 

Minor suggestions: 

Page 1, line 39: Please reformulate the sentence to avoid repetition of “and”: “In Europe, the first ones date back to the mid-twentieth century and emerged in Great Britain and were very solid in Belgium and the Netherlands.”

Page 2, line 62: Correct “they can promoting” 

Page 7, line 257-258: The first sentence is the same as in page 4, line 151-153, I suggest to reformulate it. 
